# Hydraulic Activity and Microstructure Analysis of High-Titanium Slag

**DOI:** 10.3390/ma13051239

**Published:** 2020-03-09

**Authors:** Xinkai Hou, Dan Wang, Yiming Shi, Haitao Guo, Yingying He

**Affiliations:** 1College of Materials Science and Engineering, Xi’an University of Architecture and Technology, Xi’an 710055, China; wangdanhhh@163.com (D.W.); shiyiming2019@163.com (Y.S.); ghtsea@126.com (H.G.); 2School of Chemistry and Chemical Engineering, Xi’an University of Architecture and Technology, Xi’an 710055, China; heyy@xauat.edu.cn

**Keywords:** high-titanium slag, glass phase, mineral phase, microstructure, Raman spectroscopy

## Abstract

To explain the relationship between the hydration activity of high-titanium slag and its microstructure, the hydration activity of high-titanium slag was determined, then the mineral phase and microstructure characteristics of high-titanium slag glass phase and blast furnace slag were investigated using a series of analytical methods, which contain X-Ray Diffraction (XRD), Scanning Electronic Microscope (SEM), Fourier Transform Infrared spectroscopy (FTIR), Raman spectroscopy, and Nuclear Magnetic Resonance spectroscopy (NMR). The results showed that in slow-cooled high-titanium slag, the hydration inert mineral content was about 98%, and the glass phase content was less than 2%, hence, the hydration activity of slow-cooled high titanium slag accounted for less than 25% of that of the blast furnace slag. The content of the glass phase in water-quenched high-titanium slag was 98%, but the microstructure of the glass phase was very different from that of the blast furnace slag. The glass phase of high-titanium slag has stable forms, which are TiO_4_^4−^ monomers, chain or sheet units O–Ti–O, and a small amount of 6-coordination Ti^4+^. The Ti makes the SiO_4_ tetrahedron in a glass phase network not only a monosilicate, but more stable forms of disilicates and chain middle groups also appeared. The relative bridge oxygen number increased to 0.2, hence, the hydration activity of water-quenched high-titanium slag took up less than 37% of that of the blast furnace slag.

## 1. Introduction

High-titanium slag is a kind of granular or massive waste slag that is produced by the quenching or natural cooling of molten slag discharged from the pig iron obtained via smelting vanadium titanium magnetite [1]. Hydraulicity refers to the property that a material can be hardened in humid air and water to form a stable compound after being ground into a fine powder and mixed with water to form a slurry, also known as hydraulic activity, which is the most basic property of cementitious material. Because the content of TiO_2_ in high-titanium slag is about 20%, and because the slag has low latent hydraulic activity (simply termed as hydraulic activity, also known as pozzolanic activity), it cannot be widely used as a building material. Therefore, it is stored in slag yards, thus occupying a lot of land and causing the waste of titanium resources [2]. At present, 80 million tons of high-titanium slag have accumulated in China, and it is still increasing at a rate of 3 million tons per year, and the utilization rate is less than 3% per year [3,4,5,6,7]. Chemical and mechanical activation have been used to improve the hydraulic activity of high-titanium slag, but these methods have not had a significant effect. Su et al. compared the effects of several alkali activators. When the mix proportion of the high-titanium slag was 70% in the cementitious materials, and the mix proportion of the activator was 4% in the cementitious materials, the best result increased the compressive strength of the cement mortar after 28 d from 6.5 MPa to 8.0 MPa, which is an increase of only 1.5 MPa [8]. Yang et al. studied the characteristics of grinding activated high-titanium slag. When the mix proportion of the high-titanium slag was 30% in the cementitious materials, and its specific surface area was in the range of 300–500 m^2^/kg, the compressive strength of cement after 28 d did not increase monotonically with the fineness, but there is a suitable fineness value of 400 m^2^/kg for the highest compressive strength. The compressive strength of the cement after 28 d was only 23 MPa, which still indicates a very low hydraulic activity [9].

The hydration characteristics of slag are closely related to its hydraulicity. Ao compared the hydration products of blast furnace slag and high-titanium slag, and found that there was no significant difference in the types and morphology of the hydration products of the two kinds of slag cement, except that the amount of high-titanium slag hydration products was small and the C–S–H gel crystallinity was poor and had an irregular rolled shape, and the strength of the cement specimen was lower due to lower density [10]. Shi et al. studied the hydration behavior and hydration products of crystalline minerals in slow-cooled high-titanium slag. It was thought that the existence of elemental Ti promoted the formation of hydration inert minerals such as perovskite and titanopyroxene, which is the main reason for the low hydraulic activity of the slow-cooled high-titanium slag [11]. In fact, even after water quenching treatment, the hydraulic activity of high-titanium slag was still lower and the setting time longer than that of blast furnace slag. The content and microstructure of the glass phase were the key factors affecting the hydraulic activity of slag [12,13,14]. What is the content of glass phase in high-titanium slag? Does the low content of glass phase restrict the hydraulic activity of high-titanium slag? What are the differences between the glass phase microstructure of high-titanium slag and that of blast furnace slag, and how do these differences affect the hydraulic activity of the glass phase?

In this study, the hydraulic activities of high-titanium slag and blast furnace slag are compared. Then, the mineral phase characteristics and the microstructure characteristics of the glass phase of high-titanium slag and blast furnace slag are investigated contrastively, using a series of analytical methods such as XRD, SEM, FTIR spectroscopy, Raman spectroscopy, and NMR spectroscopy. The inevitable connection between the glass phase microstructure and the hydraulic activity is also revealed, with a view to providing new theoretical guidance for improving the hydraulic activity of high-titanium slag.

## 2. Experiment

### 2.1. Raw Materials

Three kinds of slag were used in the experiment. The slow-cooled high-titanium slag was procured from Panzhihua Iron and Steel Corporation (Panzhihua, China) and was labeled TM. The water-quenched high-titanium slag was procured from Tranvic Group (Chengdu, China) and was labeled TS. In addition, the blast furnace slag was procured from Beijing Shougang Group(Beijing, China), and was labeled PS. The chemical composition (mass fraction) of the three types of slag was determined according to GB/T 176—2017, the measurement results were within the error range specified in this standard, listed in Table 1.

### 2.2. Instruments

An X-ray diffractometer (D/Max 2200, Japan, Cu Kα ray with λ = 0.15418 nm) was used for X-ray diffraction. A field emission scanning electron microscope (Hitachi S4800, Tokyo, Japan) equipped with an energy dispersive X-ray spectrometer (EDS) was used in this study. In addition, a micro-infrared spectrometer (Bruker VERTEX70 from Cernet Co., Ltd. Germany), having a frequency band range of 400 to 4000 cm^−1^ and a resolution of 2 cm^−1^, was used for infrared spectroscopy. A laser Raman spectrometer (Horiba HR Evolution, Japan) was used for Raman spectroscopy. The experiment was conducted at room temperature with a laser wavelength of 532 nm and within a frequency band of 100–2000 cm^−1^. Moreover, a fully digital NMR spectrometer (AVANCE 400 (SB) from Bruker Bio-spin, Switzerland) was used for NMR spectroscopy. The resonance frequencies of ^29^Si and ^27^Al were 79.49 Hz and 104.23 Hz, respectively.

### 2.3. Test Method

#### 2.3.1. Hydraulic Activity Test

(1) According to Appendix A of GB/T 18046—2017, the compressive strength ratio of the prepared 50% slag powder sample and Portland cement reference samples was taken as the index of hydraulic activity *A* of their age. (2) The content of active SiO_2_ and Al_2_O_3_ dissolved in saturated limewater was determined by the method of boiling circumfluence [15]. The percentage of the amount of SiO_2_ and Al_2_O_3_ dissolved to the total amount of SiO_2_ and Al_2_O_3_ is the pozzolanic activity rate *K*_a_ of the slag.

#### 2.3.2. Content of Glass Phase 

(1) According to Appendix C of GB/T 18046—2017, Jade software was used to fit the XRD patterns of the slag. The ratio of the crystal diffraction peak area to the total diffraction peak area is slag crystallinity, and 100% minus the crystallinity is the glass phase content. (2) The alkali–acid two-stage dissolution method [16] was used to dissolve the glass phase and f-CaO in the slag. The glass phase content was obtained by subtracting the f-CaO content from the mass reduction rate.

## 3. Results and Discussion

### 3.1. Hydraulic Activity Test

Table 2 shows the hydraulic activity test results of the three kinds of slags, in which *A*_7_ and *A*_28_ are the indexes of hydraulic activity for 7 d and 28 d, respectively. It can be seen from Table 2 that the 7 d indexes of hydraulic activity of the two kinds of high-titanium slag are less than 30% of that of blast furnace slag, and the 28 d indexes of hydraulic activity are less than 38% of that of blast furnace slag, so the hydraulic activity of high-titanium slag is far lower than that of blast furnace slag. Comparing the two kinds of high-titanium slag, the index of hydraulic activity for 7 d and 28 d of water-quenched high-titanium slag is 1.7 and 1.5 times that of slow-cooled high-titanium slag, respectively. The hydraulic activity of water-quenched high-titanium slag is obviously higher than that of slow-cooled high-titanium slag. *K*_a_ is the pozzolanic activity rate of slag. The order of the pozzolanic activity rate and the index of hydraulic activity of the three kinds of slag are the same: blast furnace slag > water-quenched high-titanium slag > slow-cooled high-titanium slag. The pozzolanic activity rate of water-quenched high-titanium slag is 57% of that of blast furnace slag, while the index of hydraulic activity of water-quenched high-titanium slag is less than 38% of that of blast furnace slag. Referring to Table 1, it can be seen that the active SiO_2_ and Al_2_O_3_ in high-titanium slag not only have a low dissolution rate, but also have a lower content of SiO_2_ and Al_2_O_3_ than blast furnace slag, due to the high content of TiO_2_.

### 3.2. Mineral Phase Analysis 

#### 3.2.1. Mineral Phase Identification

Figure 1 shows the XRD patterns of three kinds of slag, all of which were thoroughly dried and ground to less than 5 μm. There are broad scattering peaks of glass phase between 21° and 37° in the PS and TS patterns that belong to water-quenched slag. The results show that whether it is blast furnace slag or high-titanium slag, if it is water-quenched, the main mineral composition is glass phase. A small amount of the crystalline phases in blast furnace slag contain gehlenite (2CaO·Al_2_O_3_·SiO_2_) and akermanite (2CaO·MgO·2SiO_2_); there is no gehlenite or akermanite phase, and only a small amount of perovskite in water-quenched high-titanium slag. In slow-cooled high-titanium slag, there are almost no broad scattering peaks of the glass phase, most of which are crystal minerals. In addition to perovskite, there is also Ti-bearing diopside, anorthose, and iron. As TiO_2_ is a good nucleating agent, the high-titanium slag contains perovskite, no matter which cooling method we use.

The SEM images of two kinds of high-titanium slag under backscattered electrons are shown in Figure 2. Figure 2a shows the micro-morphology of the mineral phase of water-quenched high-titanium slag. The uniform gray region 1 accounts for more than 95% of the whole field of view. The chemical composition of the micro-region as determined by EDS is Ca_0.97_SiMg_0.49_Al_0.61_Ti_0.42_O_4.9_, which is the glass phase in the high-titanium slag. Corresponding to the broad scattering peaks of glass phase in the above XRD patterns, the glass phase is the main mineral phase in water-quenched high-titanium slag. In the gray-white area 2 in Figure 2a, the chemical composition of the micro-region is CaTiO_3_, which belongs to perovskite in high-titanium slag. The maximum grain length is about 100 μm. The large grains are mostly irregular in shape, otherwise, there are a large number of grains of about 10 μm with a dispersed distribution.

Figure 2b shows the micro-morphology of the mineral phase of slow-cooled high-titanium slag in which there is more perovskite content than Figure 2a, and the degree of enrichment of mineral particles increases. There are many 20–50 μm perovskite grains, but no 10 μm perovskite grains. In the visual field, the matrix is the light gray area 3, and the chemical composition of the micro-region is CaMg_0.62_Si_1.6_Ti_0.31_O_5.21_, which belongs to Ti-bearing diopside in high-titanium slag as another major mineral of slow-cooled high-titanium slag. In the few dark gray areas marked as 4 in Figure 2b, the chemical composition of the micro-region is Na_0.45_K_0.31_AlSi_1.19_O_4.26_. The mineral belongs to anorthose in high-titanium slag, the grain size is about 20–50 μm, and a pile of the grains is embedded in Ti-bearing diopside. In a few bright areas 5, the chemical composition of the micro-region is Fe. It is a residual metallic iron phase in high-titanium slag, with a grain size of approximately 5 to 15 μm and a round or long shape.

#### 3.2.2. Distribution Characteristics of Elemental Ti 

Figure 3 is a surface image of the Ti distribution in high-titanium slag, where (a_1_) and (a_2_) correspond to water-quenched high-titanium slag, (b_1_) and (b_2_) correspond to slow-cooled high-titanium slag. In water-quenched high-titanium slag, Ti is uniformly dispersed throughout the glass phase matrix, and the content of Ti in perovskite is higher than that of glass phase. In slow-cooled high-titanium slag, the distribution of Ti is uneven, and the anorthose and iron phases do not contain Ti. The matrix mineral, Ti-bearing diopside, has a low Ti content; the perovskite contains the highest Ti content.

#### 3.2.3. Glass Phase Content

In order to ensure the accuracy of the test results, the XRD method and the alkali–acid two-stage dissolution method were used to measure the glass phase of three kinds of slag. The test results are shown in Table 3.

As can be seen from Table 3, the glass phase content of the slow-cooled high-titanium slag is quite low. The glass phase content of the two kinds of water-quenched slag is about 98%, and the results are similar. The presence of more visible perovskites in Figure 2; Figure 3 is shown intentionally to show the morphology of the perovskites, although this does not mean the perovskite content in water-quenched high-titanium slag is high. From this point of view, the hydraulic activity of the two kinds of water-quenched slag is quite different, and it is not caused by the difference in the glass phase content. The microstructure of the two kinds of glass phase should be analyzed.

### 3.3. Analysis of Glass Phase Microstructure

#### 3.3.1. FTIR Spectroscopy Analysis 

Figure 4 shows the comparative FTIR spectroscopy of the two kinds of water-quenched slag. It can be seen from the Figure 4 that the effective spectral range of 400–1200 cm^−1^ is composed of three regions: 800–1000 cm^−1^ with strong intensity, 400–600 cm^-1^ with medium intensity, and 600–700 cm^−1^ with weak intensity. The 800–1000 cm^−1^ band is produced by the antisymmetric stretching vibration of the Si–O bond in the SiO_4_ tetrahedron in slag [17]. Compared with blast furnace slag, the peak position of water-quenched high-titanium slag here is shifted toward a higher wave number. The 400–600 cm^−1^ band is produced by the symmetric bending vibration of the Si–O–Si bond. Compared with blast furnace slag, the peak position of water-quenched high-titanium slag here is shifted toward a lower wavenumber. The 600–700 cm^−1^ band is produced by the symmetric stretching vibration of the Si–O bond [18]. There was no significant change in the peak position of this band in the two kinds of slag. Sun et al. thought that the 800–1000 cm^−1^ band shifted in the high wave number direction, and the 400–600 cm^−1^ band shifted toward the low wave number direction, indicating that the degree of polymerization of the glass phase had increased [17]. Therefore, compared with blast furnace slag, the glass phase polymerization degree of water-quenched high-titanium slag increased and the hydraulic activity decreased.

#### 3.3.2. Raman Spectrum Analysis 

Generally, the higher the molecular polarizability, the stronger the Raman spectrum intensity. Because of the high polarizability of Ti–O, Raman spectroscopy is more effective than FTIR spectroscopy for detecting the chemical environment of Ti.

The effective envelope of the Raman spectroscopy of the two kinds of slag glass phase is in the range of 500–1200 cm^−1^ region. Blast furnace slag has a small Raman shift peak at 870 cm^-1^ and the water-quenched high-titanium slag has a large packet Raman shift peak at 800 cm^-1^. In order to analyze the information provided by the overlapping peaks, the integrated envelopes in the original Raman spectrum of the water-quenched high-titanium slag were divided into six Raman shift peaks using Origin software, as shown in Figure 5.

In the silicate glass phase system, the structural units of the SiO_4_ tetrahedron can be divided into *Q*^0^, *Q*^1^, *Q*^2^, *Q*^3^, and *Q*^4^ according to the coordination number of the bridge oxygen around Si. Here, *n* in *Q^n^* represents the coordination number of the bridge oxygen around Si. The lower the bridge oxygen coordination number, the higher the hydraulic activity. The Raman shift for blast furnace slag at 870 cm^−1^ was produced by the stretching vibration of the Si–O bond in *Q*^0^ [19,20]. The integrated envelopes for water-quenched high-titanium slag were divided into six Raman shift peaks. Of these Raman shift peaks, the Raman shift at 792 cm^−1^ is generated by the Ti–O^2−^ stretching vibration in TiO_4_^4−^ monomers, the Raman shift at 724 cm^−1^ is generated by the deformation of O–Ti–O in chain or sheet units [21,22]. The Raman shift at 649 cm^−1^ is generated by Ti–O stretching vibrations in 6-coordinated Ti^4+^ [23]. The above three Raman shifts indicate that the Ti in the water-quenched high-titanium slag mainly exists as TiO_4_^4-^ monomers, the secondary form is O–Ti–O in chain or sheet units, and a small amount of Ti exists in the 6-coordinated form. The Raman shifts at 855 cm^-1^, 915 cm^-1^, and 981 cm^-1^ are generated by the stretching vibration of the Si–O bond in *Q*^0^, *Q*^1^, and *Q*^2^ [24,25,26], respectively. The results show *Q*^0^, *Q*^1^, and *Q*^2^ structural units of the SiO_4_ tetrahedron in water-quenched high-titanium slag.

It can be seen from the above test results that the SiO_4_ tetrahedron in blast furnace slag is mainly monosilicate represented by *Q*^0^, so its hydraulic activity is high. TiO_4_^4-^ monomers in water-quenched high-titanium slag are more stable than the SiO_4_ tetrahedron [23]. At the same time, there is O–Ti–O in chain or sheet units, which transform the original non-bridged oxygen to bridged oxygen, so that the number of bridged oxygen per SiO_4_ tetrahedron increases, and the degree of network polymerization is enhanced. The Ti in water-quenched high-titanium slag enters the silicate glass system in the form of a network formers, which not only reduces the content of the SiO_4_ tetrahedron, but also transforms some of the SiO_4_ tetrahedron to the more stable form of *Q*^1^ disilicates and *Q*^2^ chain middle groups. Therefore, the glass phase structure of water-quenched high-titanium slag is more stable than that of blast furnace slag, and its hydraulic activity is lower than that.

#### 3.3.3. NMR Analysis

(1) ^29^Si spectrum and ^27^Al chemical environment

The relationship between chemical shift and structure in ^29^Si NMR is shown in Table 4. Al mainly exists in the form of four-coordination or six-coordination in the glass phase network in aluminosilicate. The chemical shift between +50 and +80 ppm belongs to the [AlO_4_] tetrahedral structure, and the chemical shift between –10 and +20 ppm belongs to the [AlO_6_] hexahedral structure [27].

Figure 6 shows the ^29^Si and ^27^Al NMR spectroscopy for two kinds of water-quenched slag, and Figure 6a shows the NMR spectrum of ^29^Si. It can be seen from Figure 6a that there is only *Q*^0^ peak in blast furnace slag, indicating that the SiO_4_ tetrahedron in blast furnace slag only exists in the form of monosilicate. Relative to blast furnace slag, the chemical shift of water-quenched high-titanium slag moves to the high field, and the peak is not a *Q*^0^ single peak, but the superposition peak of *Q*^0^ and *Q*^1^. This shows that the SiO_4_ tetrahedron in water-quenched high-titanium slag exists in the form of monosilicate and disilicates, which is consistent with the analysis of FTIR spectroscopy and Raman spectroscopy. Figure 6b shows the NMR spectrum of ^27^Al. The ^27^Al peaks of blast furnace slag and water-quenched high-titanium slag are 65.5 ppm and 62.8 ppm, respectively, so the chemical environment of Al element in the two kinds of slag is [AlO_4_] tetrahedral, and the presence of Ti has no obvious influence on the structure of Al.

(2) Characterization of polymerization degree

The polymerization degree is an index reflecting the comprehensive influence of the chemical composition and glass phase structure on the hydraulic activity. The higher the polymerization degree, the lower the hydraulic activity. The change in the bridging oxygen number can be used to reflect the polymerization degree of the system. The bridging oxygen number increases with Si–O bond polymerization and decreases with Si–O–Si bond depolymerization. Therefore, the higher the bridging oxygen number, the higher the degree of polymerization and the lower the hydraulic activity.

In this study, the deconvolution of the ^29^Si NMR spectroscopy of TS was performed using Origin software, and the relative areas of the resonance peaks were calculated. The magnitude of the corresponding relative bridging oxygen (RBO) number was calculated according to the following formula [29]:(1)$$\mathrm{RBO}=\frac{1}{4}(1\times \frac{Q^{1}}{\sum Q^{n}}+2\times \frac{Q^{2}}{\sum Q^{n}}+3\times \frac{Q^{3}}{\sum Q^{n}}+4\times \frac{Q^{4}}{\sum Q^{n}}),$$where *Q^n^* is the relative area of the corresponding resonance peak.

As shown in Figure 7, the ^29^Si NMR spectroscopy of water-quenched high-titanium slag is deconvoluted into five peaks. Based on classification according to the chemical shift, there are two *Q*^1^ peaks, two *Q*^0^ peaks, and one *Q*^2^ peak. Therefore, the *Q*^1^, *Q*^0^, *Q*^2^ forms of the SiO_4_ tetrahedron exist in the water-quenched high-titanium slag. Because *Q*^1^ has the largest area, it can be said that disilicate is the main form in water-quenched high-titanium slag. After calculating the integral area of each peak, the relative areas of *Q*^1^, *Q*^0^, and *Q*^2^ were found to be 100, 81.06, and 21.82, respectively, and the calculated RBO value was 0.2. Blast furnace slag has only one *Q*^0^ single peak, consequently, its RBO was 0. Therefore, the high degree of polymerization and more stable structure of the glass phase in water-quenched high-titanium slag are the main reasons for its low hydraulic activity.

## 4. Conclusions

The main mineral phase composition of the slow-cooled high-titanium slag included perovskite, Ti-bearing diopside, and anorthose. Because these mineral phases pertain to inert mineral crystals, and the content of glass phase was less than 2%, the hydration activity was low. Although the content of glass phase in water-quenched high-titanium slag was 98%, Ti mainly existed in the glass phase structure in the form of TiO_4_^4-^ monomers, chain or sheet units, and a small amount of 6-coordination, which were relatively stable. The titanium oxygen structure in the glass phase not only reduced the relative content of SiO_4_ tetrahedron, but also made the SiO_4_ tetrahedron in the glass phase network a monosilicate, and more stable forms of disilicates and chain middle groups appeared. The relative bridge oxygen number increased to 0.2, so the hydraulic activity of the water-quenched high-titanium slag was also low. Based on the results of this study, the removal of Ti from high-titanium slag not only increased the relative content of active SiO_2_ and Al_2_O_3_ in the glass phase, but also improved the hydraulic activity of the glass phase.

## Figures and Tables

**Figure 1 materials-13-01239-f001:**
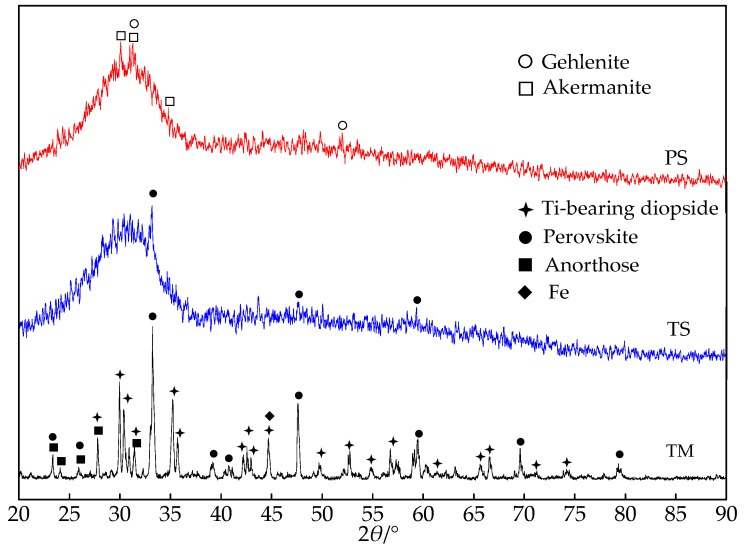
XRD patterns of three kinds of slag.

**Figure 2 materials-13-01239-f002:**
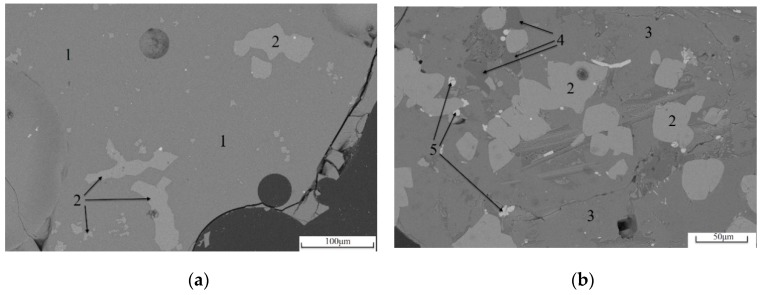
SEM images of two kinds of high-titanium slag. (**a**) TS; (**b**) TM.

**Figure 3 materials-13-01239-f003:**
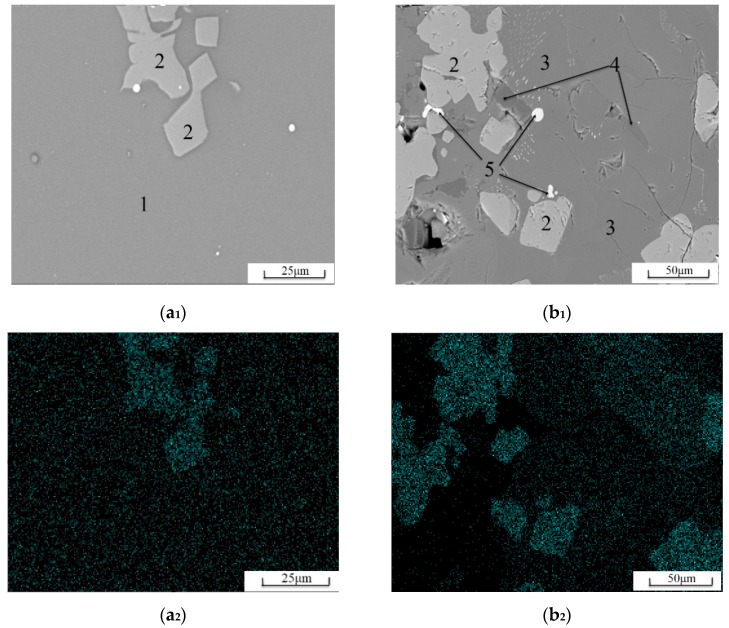
SEM images of Ti surface scanning of high-titanium slag. (**a_1_**) SEM image of TS; (**a_2_**) Ti surface scanning of TS; (**b_1_**) SEM image of TM; (**b_2_**) Ti surface scanning of TM.

**Figure 4 materials-13-01239-f004:**
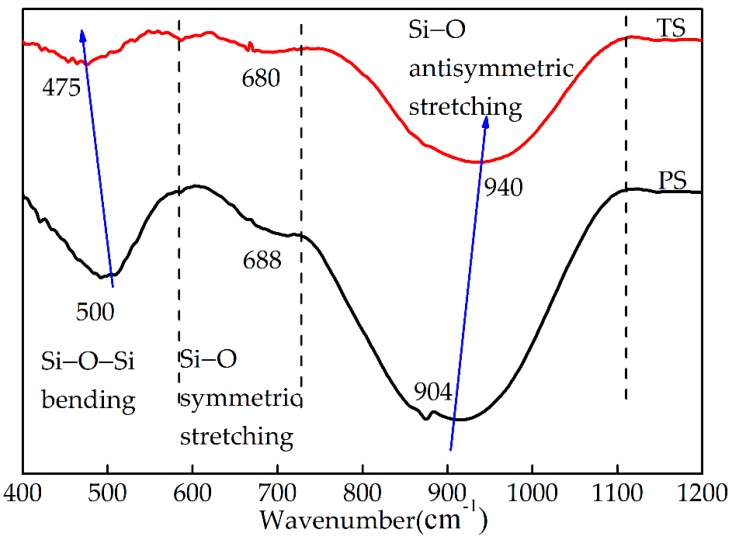
FTIR spectroscopy of two kinds of water-quenched slag.

**Figure 5 materials-13-01239-f005:**
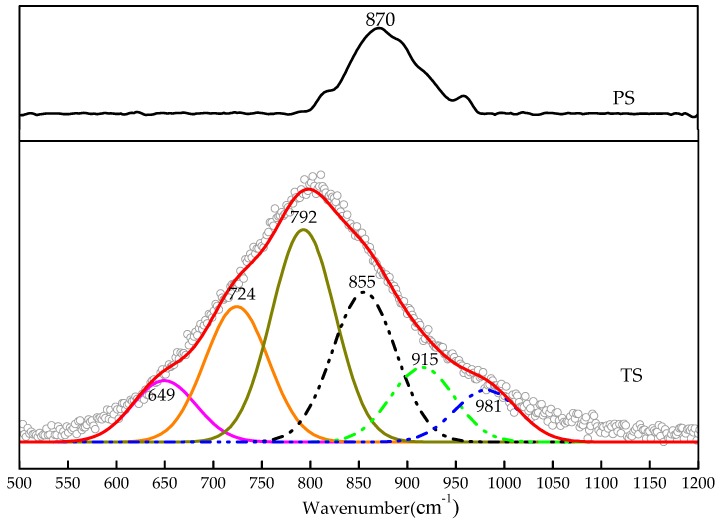
Raman spectroscopy of two kinds of water-quenched slag.

**Figure 6 materials-13-01239-f006:**
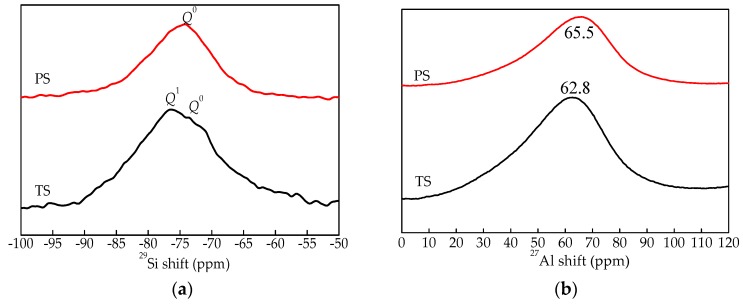
^29^Si and ^27^Al NMR spectroscopy for two kinds of water-quenched slag. (**a**) ^29^Si NMR spectroscopy; (**b**) ^27^Al NMR spectroscopy.

**Figure 7 materials-13-01239-f007:**
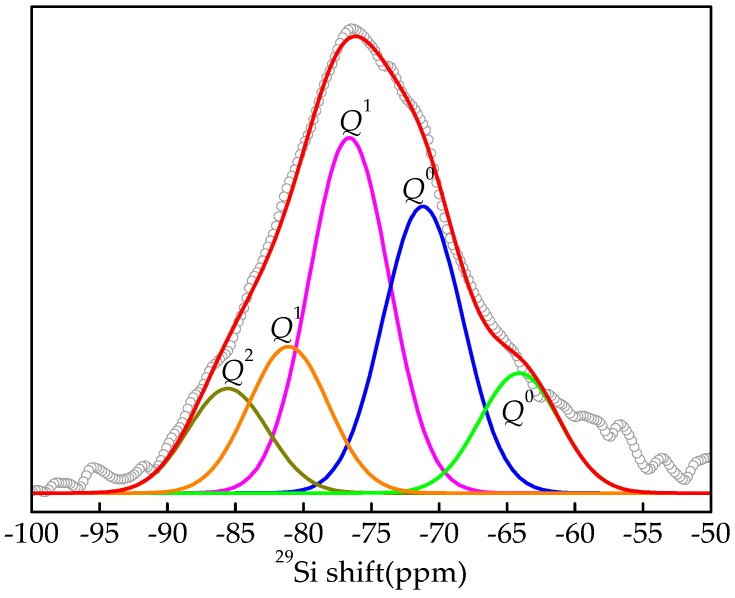
Deconvolution of ^29^Si NMR spectroscopy of water-quenched high-titanium slag.

**Table 1 materials-13-01239-t001:** Chemical composition of slag (wt%).

Sample	CaO	SiO_2_	Al_2_O_3_	TiO_2_	MgO	Fe_2_O_3_	Na_2_O	LOI	Sum
TM	26.64	24.76	13.22	20.39	8.37	1.20	0.95	1.64	97.17
TS	28.36	26.36	13.33	17.18	8.39	1.31	0.16	1.30	96.39
PS	35.92	33.65	16.90	0.07	10.13	0.13	0.33	1.55	98.67

**Table 2 materials-13-01239-t002:** Hydraulic activity test of slag (%).

Sample	*A* _7_	*A* _28_	*K* _a_
PS	125.6	95.6	99.3
TS	35.1	35.8	56.8
TM	21.2	23.5	35.7

**Table 3 materials-13-01239-t003:** Comparison of glass phase content in slag (%).

Method	PS	TS	TM
XRD	98.03	97.34	1.82
alkali-acid two stage dissolution	99.49	98.35	1.91

**Table 4 materials-13-01239-t004:** ^29^Si NMR chemical shifts of the structure unit of *Q^n^* in solid silicates [28].

Types of Si-O-X group	Symbol	Chemical Shift/ppm
Monosilicate	*Q* ^0^	−68–−76
Disilicates and chain end groups	*Q* ^1^	−76–−82
Chain middle groups	*Q* ^2^	−82–−88
Layers and chain branching sites	*Q* ^3^	−88–−98
Three-dimensional networks	*Q* ^4^	−98–−129

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
