# Peer review of "Hydraulic Activity and Microstructure Analysis of High-Titanium Slag"

_materials, 2020, doi:10.3390/ma13051239_

Round 1
Reviewer 1 Report
Dear Author,
A few questions and suggestions to the author are given below:
In Table 1, where the chemical compositions of three different slags are given, LOI is given, it would be good to give the expansion. How did the chemical composition given determine, please mention the instrument or analysis, or if it is already available, please give the source as reference? How did you determine the chemical composition, was it a single time measurement, If not please include the error in the compositions. The content of active SiO2 and Al2O3 dissolved in saturated limewater was determined by the method of boiling circumfluence. Was the slag completely soluble. If soluble completely, why didn’t the author try ICP or AAS ? Please give more clear description, since A7 and A 28 are mentioned in table 2 The XRD given are very broad showing the amorphous nature. Are these dry samples or the wet sample? If dry please mention the condition. If not dried , please dry the sample and record the XRD for a better identification. How did the glass phase content determine? What is alkali –acid two stage dissolution method, give reference or give a description.Overall the manuscript needs more clarity. So, I recommend the acceptance of the manuscript after rewriting the manuscript.
Reviewer 2 Report
The manuscript "Hydraulic activity and microstructure of high-Titanium slag" gives a nice comparison of the hydraulic activity of three different kinds of slags. Structure and hydraulic activity are correlated with each other carefully using different techniques. I have only a few comments which should be adressed to make the results very clear.
(1) Explain in the introduction what hydralic activity/Hydration acitivity is. Why is it imporant? It is not clear for a reader who is not from the field.
(2) Why is the pozzolanic activity rate of the slag important?
(3) Please give composition/structural formulas for the detected Minerals.
(4) I would always prefer Raman "shift" instead of "Peak".
After editing I strongly recommend publication of the manuscript.
Reviewer 3 Report
Please place the figures after they have been discussed. It is weird that you provide the figures first, then explain it. For example, section 3.2.1 starts with a figure!! Similarly, section 3.3.1 starts with a figure.
Some grammatical errors can be seen throughout the manuscript. A quick review will catch them.
Reviewer 4 Report
The manuscript entitled “Hydraulic activity and microstructure analysis of high-titanium slag” revealed the relationship between the hydration activity of high-titanium slag and its microstructure. The manuscript presents experimental work to investigate the mineral phase and microstructure characteristics of high-titanium slag glass phase and blast furnace slag.
The manuscript is good in quality and information. Therefore, this reviewer recommends accepting it after considering the following comments.
Technical Comments:
- The manuscript could benefit greatly from professional editing to improve technical writing and English.
- Lines 10-11: It is a very poor sentence to start with. The author should mention an answer to the sentence " In order to reveal the relationship between the hydration activity of high-titanium slag and its microstructure". Determination of the hydraulic activity is not an answer to reveal the relationship.
- Line 13: What you mean by "such as"? Do you mean that you used many and you mentioned a few examples? Also, why all of these methods used to investigate the mineral phase and microstructure characteristics?
- Line 17: What do you mean by "low"? Low is a general word. I can say one million is low relative to one milliard. Comparisons with another case will be better.
- Line 64: The authors should increase their discussion on previous related research and highlight how their study is providing a different approach or adding significantly to what has been done.
- Line 108: Do you mean 7 days and 28 days?
- Lines 279-281: It is a too long and more complicated sentence. Can you rephrase it to be more readable?
Round 2
Reviewer 1 Report
The authors have given a detailed reply to the querries and have incorporated the suggestions. The manuscript can now be accepted.
